# Carbon metabolic rates and GHG emissions in different wetland types of the Ebro Delta

**Daniel Morant**[1], **Antonio Picazo**[1], **Carlos Rochera**[1], **Anna C. Santamans**[1], **Javier Miralles-Lorenzo**[1], **Alba Camacho-Santamans**[1], **Carles Ibañez**[2], **Maite Martínez-Eixarch**[2], **Antonio Camacho**[1]*

**1** Cavanilles Institute for Biodiversity and Evolutionary Biology, University of Valencia, Paterna, Spain,
**2** IRTA - Institute of Agrifood Research and Technology, Sant Carles de la Ràpita, Spain

* antonio.camacho@uv.es

**Data Availability Statement:** All sequence data from our study were deposited in the Sequence Read Archive (SRA) of the National Centre for Biotechnology Information (NCBI) and have been made fully accessible with the BioProject accession

## Abstract

Deltaic wetlands are highly productive ecosystems, which characteristically can act as C-sinks. However, they are among the most threatened ecosystems, being very vulnerable to global change, and require special attention towards its conservation. Knowing their climate change mitigating potential, conservation measures should also be oriented with a climatic approach, to strengthen their regulatory services. In this work we studied the carbon biogeo-chemistry and the specific relevance of certain microbial guilds on carbon metabolisms of the three main types of deltaic wetlands located in the Ebro Delta, north-eastern Spain, as well as how they deal with human pressures and climate change effects. We estimated the metabolic rates of the main carbon-related metabolisms (primary production and respiration) and the resulting carbon and global warming potential balances in sites with a different salinity range and trophic status. With the results obtained, we tried to define the influence of possible changes in salinity and trophic level linked to the main impacts currently threatening deltaic wetlands, on the C-metabolisms and GHG emissions, for a better understanding of the mitigating capacity and their possible enhancement when applying specific management actions. Metabolic rates showed a pattern highly influenced by the salinity range and nutrients inputs. Freshwater and brackish wetlands, with higher nutrient inputs from agricultural runoff, showed higher C-capture capacity (around 220–250 g C m$^{-2}$ y$^{-1}$), but also higher rates of degradative metabolisms (aerobic respiration and $CH_4$ emissions). Contrastingly, the rates of C-related metabolisms and C-retention of *Salicornia*-type coastal salt marshes were lower (42 g C m$^{-2}$ y$^{-1}$). The study of the microbial metacommunity composition by the16S RNA gene sequencing revealed a significant higher presence of methanogens in the salt marsh, and also higher metabolic potential, where there was significantly more organic matter content in sediment. Salinity inhibition, however, explained the lower respiration rates, both aerobic and anaerobic, and prevented higher rates of methanogenesis despite the major presence of methanogens. Conservation measures for these wetlands would require, overall, maintaining the sediment contributions of the river basin intending to overcome the regression of the Delta and its salt marshes in a climate change scenario. Particularly, for reducing degradative metabolisms, and favour C-retention, nutrient inputs should be controlled in freshwater and brackish wetlands in order to reduce eutrophication.

number PRJNA595160 (https://www.ncbi.nlm.nih.
gov/Traces/study/?acc=PRJNA595160).

**Funding:** This work was supported by the project
CLIMAWET (CGL2015-69557-R), funded by the
MINECO (Agencia Estatal de Investigación) of the
Spanish Government, and by FEDER-EU Funds,
awarded to AC. http://www.ciencia.gob.es/portal/
site/MICINN/aei This work was additionally
supported by the projects CARBONSINK and
CARBONNAT funded by Fundación Biodiversidad,
also awarded to AC. https://fundacion-
biodiversidad.es/ DM and JM-L hold a FPU
Predoctoral Scholarship by the Spanish Ministry of
Science, Innovation and Universities (FPU16/
01444 and FPU15/03930). http://www.ciencia.gob.
es/.

**Competing interests:** The authors have declared
that no competing interests exist.

In salt marshes, the reduction of salinity should be avoided to control increases in methano-
genesis and $CH_4$ emissions.

## Introduction

Wetlands located in deltaic environments are highly diverse, as they are located in transitional
areas between continental, riverine and marine conditions [1–3]. They are highly dependent
on the river supplies of freshwater sediments, which are very important to maintain their par-
ticularities confronting the marine influence [4–6]. These deltaic areas support high levels of
biodiversity and generate a wide variety of ecosystem services [7]. They are also among the
most productive ecosystem types [8] and usually function as natural carbon (C) sinks [9–11].
However, due to their location in highly productive areas, they experience a sort of direct and
indirect impacts that makes them among the most endangered systems due to human threats
[12]. Some of the most important impacts are caused by surrounding anthropic uses [13],
especially by agriculture [14], with consequences in the degradation of the conservation status
and a huge loss of wetlands surface [1]. In addition, deltaic areas are especially vulnerable to
the ongoing climate change, being especially sensitive to sea level rise [13].

Currently many river deltas are under regression caused by both the reduced riverine sedi-
ment supply as retained by dams in regulated rivers, as well as by the sea level rise. Addition-
ally, the loss of deltaic wetland surface in favour of anthropic uses worldwide [15], and the
incidence of other threats in synergy with climate change [16], requires an effective strategy
for the conservation of these valuable ecosystems. However, both threats as the above men-
tioned, but also the possible restoration and conservation measures, might greatly influence
the C-accumulation rates [17] and, in some cases, could even favour the release of the already
stored buried carbon [18]. Therefore, conservation solutions and management actions should
not only try to ensure their ecological functionality and promote the resilience of ecosystems
and their ability to adapt to changes in the environment [19], but also could aim strengthening
properties like the C-sequestering capacity [20], that can be used to fight against climate
change [21].

For a proper consideration of the potential role of deltaic wetlands as climatic allies, a first
step would be understanding the mechanisms and factors of carbon retention and GHG emis-
sion, and how they are related with the biological communities settled in the systems and with
some key ecological factors, as we afforded in this work.

Photosynthesis and respiration both in the water column and the benthos [3], and aquatic
plant productivity [8] are the major contributors to organic matter production and mineralisa-
tion. The rates of these C-related metabolisms can be linked to the ecosystem functioning and
its conservation status [18, 22]. $CH_4$ emissions, resulting from the balance between methano-
genesis and methanotrophy, can also be important in deltaic wetlands [23], although salinity, a
key factor in these coastal systems, can regulate these rates [24]. The Net Ecosystem Metabo-
lism (NEM) can be assessed by estimating the differences between the C-fixation and C-degra-
dative metabolisms [25]. From these estimations, the role of the studied wetland types in GHG
exchange with the atmosphere can be assessed. Additionally, for further understanding of the
biological mechanisms related to the C-exchange with the atmosphere, the study of the micro-
bial diversity in the sediments related to some key anaerobic respiration processes can be
added to the ecosystem metabolic study [26]. This would allow relating the association
between some specific microbial guilds and the main C-metabolic pathways, then trying to
relate all these issues to suitable management measures and potential environmental changes.

The Ebro Delta, in the westernmost part of the Mediterranean Sea, shows a maximum tidal range of around 0.25 m [27]. The sustainability of the Delta largely depends on the river basin management [28]. The Ebro river is highly regulated throughout its entire course, modifying the sediment contributions and flow regime towards the Delta and its ecosystems. Currently, the Ebro Delta is undergoing coastal retreat at the river mouth, mainly because of the major implications of wave erosion and elevation loss due to the decreases of new sediment contributions [17]. The storm occurring in January 2020 practically left the entire Delta flooded (S1 Fig), and may have resulted in the damages to many habitats. Most of the total deltaic surface, of around 330 $km^2$, is basically occupied by rice fields (65%). Apart from additional urban uses and sand beaches, several wetland types still occupy part of the Ebro Delta. Some of them are salt marshes located at the coast, whereas others are freshwater to brackish wetlands, with different plant communities, distributed along the Delta but separated from the sea [29]. Further than these two main wetland types, freshwater wetlands were also abundant in the Ebro Delta, though many of them were transformed to rice fields. From few decades ago, some freshwater wetlands have been recovered from abandoned rice fields. This third wetland type characteristic from the Ebro Delta, apart from recovering natural values, is currently acting as a biological filter to reduce nutrients concentrations from agricultural runoff before pouring the water into the sea [30]. Both freshwater and brackish wetland types are, in general, highly pressured by the surrounding crop fields, receiving mostly freshwaters through channels that come indirectly from the Ebro river, but also collecting runoff from rice fields with certain nutrient loads [31].

In this paper we tried to evaluate the main metabolic rates related to carbon retention and GHG emission, in order to determine how they can be influenced by key ecological features like the salinity and the trophic status. The metabolic capacity was also studied and related with the linkage of these emissions with the microbial diversity. We combined limnological and metabolic in situ studies and experimentation with the assessment of the relative relevance of some microbial guilds responsible of the methanogenesis. Particularly, we studied a salt marsh almost connected to the sea, and two wetlands that are not directly connected to the sea, namely a brackish shallow wetland, and a recovered freshwater wetland, submitted to pressures related to the surrounding agriculture and the Delta regression, enhanced by the effects of climate change in the area [32]. Our hypothesis was that salinity, and changes in salinity due to the delta regression and waterflow modifications, as well as the degree of trophic alteration, would influence metabolic rates. Based on the trends and degree of influence of these factors on the main carbon-related metabolisms, primary production and the most relevant types of respiration, we tried to define the possible effect of some conservation actions to favour the mitigating capacity of the main types of deltaic wetlands. Using a novel approach, in this study we jointly addressed the importance of the wetland's ecological characteristics and their relationship with their carbon metabolism, as well as how they can be modified by the pressures they are facing. From this study, management actions with a climatic focus were suggested based on our results that could be used for deltaic wetlands protection and the preservation of their ecosystem services.

## Material and methods

### Study sites

Three sites were selected as representative of the three main types of deltaic wetlands located in the Ebro Delta, (Catalonia, Spain). These were natural coastal salt marshes, natural brackish coastal wetlands, and (restored) freshwater wetlands. Salt marshes are separated from the sea just by a very narrow sand bar, whereas brackish wetlands do not have relevant surface seawater inlets but have the influence of marine intrusion determining its salinity. Freshwater

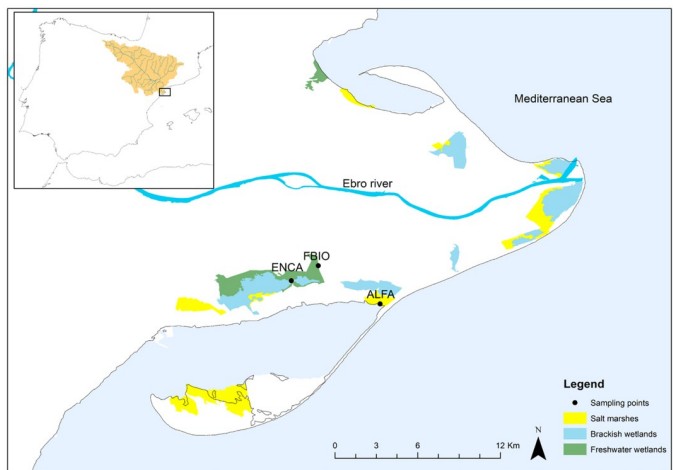

**Fig 1. Map of the location of the Ebro river basin within the Iberian Peninsula, and of the Ebro Delta, with the location of the three studied sites and the total area covered by the each of the three different wetland types studied.**

wetlands, the third studied wetland type, receive mainly freshwaters from the Ebro River after circulating through an intricate network of irrigation channels. The sites selection considered the range of salinity found in these environments, as well as the range of trophic status currently appearing among the Ebro Delta wetlands (Fig 1).

Alfacs marshes (ALFA, 40˚38'17"N; 0˚44'35"E) is a microtidal salt marsh of around 42 ha. with a mosaic of patches of open shallow waters and temporary flooded areas covered by *Salicornia* spp., the later covering around 37% of the above mentioned total area. It is separated from the Mediterranean by only a very narrow (few metres wide) sand beach. Encanyissada (ENCA, 40˚39'11"N; 0˚41'7"E) is a brackish coastal wetland which occupies more than 500 ha., with a depth controlled by agricultural water inputs and outputs, draining to the sea through artificialized channels. After its alteration and the reclamation of the surrounding area for rice paddies, formerly only drainage water from rice fields entered ENCA, whereby showed alarming levels of eutrophication. Currently, the amount of water that comes from the rice fields has remarkably decreased. However, hydrological inputs are highly variable, and conductivity can vary from around 5.4 mS cm$^{-1}$ (when the experiment was obtained), to more than 60 mS cm$^{-1}$ in some periods. In spite of this, this wetland typically presents brackish waters, as evidenced by the *Salicornia* spp. meadows extending through the wetland shores, although there are also some areas occupied by *Phragmites australis* in the areas closer to freshwater inlets. Filtre biològic (FBIO, 40˚39'47"N; 0˚42'11"E) is a wetland recovered on 43 ha. previously occupied by rice fields. Some natural values have been recovered, and this wetland currently acts as a biological filter for the agriculture runoff, reducing nutrients concentrations by natural processes. Water inputs and outputs are also controlled, maintaining a regular water depth. Helophytes, mainly *P. australis*, cover around 50% of the wetland surface, accompanied by other autochthonous helophytes like *Typha* spp., which are highly efficient in the reduction of nutrient loads.

## Limological analyses

The study covered two hydrological cycles, 2015/2016 and 2016/2017 through seasonal sampling. Permits for sampling were granted by the IRTA (Institute of Agrifood Research and

Technology), belonging to the Catalan Government, which participated in the research. Water depth remained almost constant both in ENCA (~20 cm) and FBIO (~45 cm), due to the water level regulation through the irrigation channels network. ALFA showed usually depths of around 20 cm, with higher dependence on the sea level and some peaks of higher maximum depth depending on the microtidal cycles. Water samples were obtained at approximately 15–20 cm depth. Some water variables, such as temperature (˚C), conductivity (mS cm$^{-1}$), and dissolved oxygen (DO, mg L$^{-1}$) were determined in situ using a multiparameter probe WTW Multi 3410 logger. The specific probes used were WTW Tetracon$^{®}$ 925 IDS for conductivity and WTW FDO$^{®}$ 925 IDS for oxygen and temperature. A salinity correction was automatically applied to DO measurements. The pH was measured with a Crison Basic-20 pH-meter. Other basic limnological variables were determined in the laboratory following the protocols given by APHA [33]. These were alkalinity (meq. L$^{-1}$), total suspended solids (TSS), and the concentrations of soluble orthophosphate (SRP), nitrate ($NO_3^-$) and ammonia ($NH_4^+$). Chl-*a* concentration was determined following Picazo et al [34].

Sediment features were also studied in detail in the three last sampling dates (January, April and August 2017). At each site and every sampling date, three replicates sediment cores were obtained from the first 5–10 cm depth. pH and conductivity were measured by 1/5 dilution of the sediment. Conductivity was measured with a WTW LF-191 conductivity meter, and pH was obtained with a Crison Basic-20 pH meter. Oxidation reduction potential (ORP) was measured directly in sediment with an electrode Sentix ORP-T900-P, then referred to the hydrogen potential (Eh). The organic matter and the carbonate content of the sediments (in % of dry weight) were obtained by the loss on ignition method at 460 ˚C/6 h (LOI460) and 950 ˚C/4 h (LOI950) respectively [35].

## Metabolic rates of the main carbon processes

$CO_2$ exchanges derived from photosynthesis and aerobic respiration were assessed discerning the contribution of (i) plankton (ii) benthos, and (iii) helophytes. To do this, direct primary production and (aerobic) respiration rates of plankton and benthos were obtained by measuring the oxygen variations after an incubation of 2–4 h of each compartment under 'light' and 'dark' conditions [18, 24], giving the Net Community Production (NCP) and the Respiration, respectively. Gross Primary Production (GPP) was calculated by the addition of NCP + Aerobic Respiration. For plankton assays, four clear Winkler-type and four dark bottles were incubated. The same measurements for the benthic compartment were conducted using bell-jars staked in the mud also with four clear and four dark jars. The areal gross primary production (GPP) and respiration rates in plankton were calculated from depth integration. In the case of benthos, areal rates also considered the surface covered by the jars. For GPP, results were converted to diel estimates by considering the length of the daytime at each sampling campaign. Respiration rates were assumed constant over a 24-hour period in order to obtain the diel estimates.

Production of emerged helophytes was assessed as the accumulated matter by somatic growth between sampling events by a harvesting approach. At each sampling campaign, the dry weight of helophytes was obtained from fifteen sampling quadrats (as replicates, 0.25 m$^2$ each) within the vegetated area of each studied wetland. The average dry weight of replicates was subsequently transformed to C content by applying a coefficient attending to the plant type, being 0.45 for helophytes [36], mainly *Phragmites australis*, and 0.255 for *Salicornia* spp., as an average of the C content found in these species [37–39]. Net production was then calculated as the difference between each two sampling events.

$CH_4$ emissions were estimated as a methanogenesis-methanotrophy balance by an ex situ incubation method using replicated sediment cores [24]. This method consisted in the extraction of 12 replicates (per site and sampling campaign) of transparent methacrylate tubes of 50 cm in length and 4 cm in diameter, where sediment (5–10 cm) and water (15–25 cm) were collected, leaving a layer of air over the water in the tube. The tubes were then incubated them for 2 to 5 days at field temperature on a thermostatic climatic chamber with light/dark cycle regulation. Then $CH_4$ accumulated in the air chamber of the tubes was measured with an Aeroqual Gas A200 sensor equipped with a gas-sensitive semiconductor (GSS) properly calibrated by gas chromatography [24].

A multivariate distance-based redundancy analysis (dbRDA) was performed with the seasonal samplings in the three studied sites, considering both the environmental data and the metabolic rates, to see the patterns between these features using the R programming.

To conduct the laboratory experiments to determine the response of the $CH_4$ emissions to key environmental variables, such as temperature and salinity, the method described by Camacho et al. [24] was used. Cores set as previously described were incubated into different conditions of temperature and salinity (5 replicates each). For the study of the temperature effects, they were incubated at four different temperatures (14, 20, 25, 30˚C) in climatic chambers. For the study of the salinity, some dilutions and concentrations of salt concentrations in the water were made. Specifically, for ALFA, the most saline site, the current salt concentration was varied as 1/5x, 1/2x, 1x, and 2x; for the brackish ENCA site by 1/5x, 1/2x, 1x, and 2x; whereas for the freshwater FBIO, salinity assayed conditions were 1/2x, 1x, 2x, and 5x. For the salinity experiments cores were previously pre-incubated during several days for the acclimatization to the corresponding assayed salinities under a constant temperature of 20˚C, then the air chamber was purged, and each tube sealed and incubated for several days in the conditions described before the measurement of the $CH_4$ emissions [24]. The response of $CH_4$ emissions to temperature was fitted to the best curve fitting and the statistics to assess the significance of the curve regressions were carried out using the software Curve ExpertBasic 1.4.

## Microbial communities

Sediment samples for the microbial study were taken at the same point as water samples but, as the rest of sediment samples, only for the three last sampling campaigns of the 2016–2017 hydrological cycle. Once collected, they were homogenized, aliquoted in subsamples and frozen at -80 ˚C until subsequent DNA extraction. DNA extraction from each sample was performed using the EZNA Soil DNA isolation kit (Omega Bio-Tek, Inc., Norcross, GA, USA) following the instructions given by the supplier. DNA sequences were obtained by Illumina sequencing of the 16SrRNA gene, using these sequences for the taxonomic (RDP 11.0 database) and the metabolic assignment (PICRUSt2). Sequencing of the region V4 of the 16S rRNA gene was done using the Illumina MiSeq system (2x250bp) at the genomics facilities of the Research Technology Support Facility of the Michigan State University, USA. For each sample, Illumina compatible, dual indexed amplicon libraries of the 16S-V4 rRNA hypervariable region, were created with primers 515f/806r. Completed libraries were batch normalized using Invitrogen SequalPrep DNA Normalization Plates. Then, the product recovered from the plats was pooled. The pool was QC'd and quantified using a combination of Qubit dsDNA HS, Agilent 4200 TapeStation High Sensitivity DNA and Kapa Illumina Library Quantification qPCR assays. This pool was loaded on a standard Illumina MiSeq v2 flow cell and sequencing was performed in a 2x250bp paired end format using a MiSeq v2 500 cycle reagent cartridge. Custom sequencing and index primers complementary to the 515/806 target sequences were added to appropriate wells of reagent cartridge. Base calling was done by Illumina Real Time

Analysis (RTA) v1.18.54 and output of RTA was demultiplexed and converted to FastQ format with Illumina Bcl2fastq v2.19.1. Sequences were processed using the UPARSE pipeline using USEARCH v11.0.667 [40]. After merging of read pairs, the dataset was filtered by a maximum number of expected errors of 0.5. Chimeric sequences were removed with USEARCH v11.0.667 utilizing the UCHIME [40], against the RDP 11.0 database [41].

PICRUSt2 (https://github.com/picrust/picrust2) [42] was used for functional community profiles, specifically to predict methanogenesis for filtered reads and to ascribe the taxonomic contributions to inferred functional pathways (MinPath, [43]) using the mp hidden-state prediction method [44]. All sequence data from our study have been deposited in the Sequence Read Archive (SRA) of the National Centre for Biotechnology Information (NCBI), BioProject accession number PRJNA595160.

Regression analyses and ANOVAs were performed in order to compare the organic matter content in the sediments (% LOI) with the percentages of methanogens, as well as the metabolic assignments (PICRUSt2) for acetoclastic methanogenesis. ANOVAs were also performed to compare the relative relevance of the metabolic assignments (PICRUSt2) for acetoclastic methanogenesis among the studied sites (n = 3 per site and sampling date). All these statistical analyses were performed using the vegan package version 2.5–4 [45] in the R programming environment [46].

### Carbon balance

The C-budget of the main metabolic activities in each studied wetland was assessed by resolving the mass balance of C inputs and outputs, expressed in terms of areal rates of C flux [18]. All the metabolic rates were transformed to C units according to the stoichiometry of the measured processes based on molar relationships. C inputs considered those metabolisms that assimilated C, namely plankton and benthos GPP, as well as helophytes (mainly *P. australis*) and halophytes (*Salicornia* spp.) production. C outputs were assimilated to those metabolisms that release C as $CO_2$, plankton and benthos aerobic respiration, or as $CH_4$ emissions. Being areal rates, the percentage of the surface where those metabolisms take place were considered (open waters and helophytes-halophytes coverage) for the overall calculations of each wetland and the extrapolation to the whole Ebro Delta per wetland type.

Additionally, the global warming potential balance (GWP) was determined by considering the warming capacity of the two carbon greenhouse gases (C-GHG) involved in the C-balance, $CO_2$ and $CH_4$. Thus, rates were also estimated as $CO_2$-equivalents, following the warming capacity of each C-GHG. According to the IPCC reports [47], a value of 28 was used for the $CH_4/CO_2$ relationship.

For the extrapolation from the results obtained in the studied sites to the rest of surface covered by each of these wetland types in the Ebro Delta, the surface occupied by each of the three types of studied wetlands was obtained from Corine Land-Cover [48], then extrapolations were made considering the surface covered by each wetland type.

### Results

The three selected sites were limnologically quite different, as shown in Fig 2, as well as in S1 and S2 Tables for the water and sediment features. They are all shallow wetlands, with average depths around 45, 20 and 50 cm in ALFA, ENCA and FBIO, respectively (S1 Table). The main differential feature that allows characterizing the three sites differently was the water electric conductivity, as indicating salinity (Fig 2 and S1 Table). In ALFA, average conductivity was 56.6 mS cm$^{-1}$, with maximum values of 78.8 mS cm$^{-1}$. In ENCA, however, the average conductivity of its brackish waters was 31.3 mS cm$^{-1}$, with wide ranges due to the high variability of

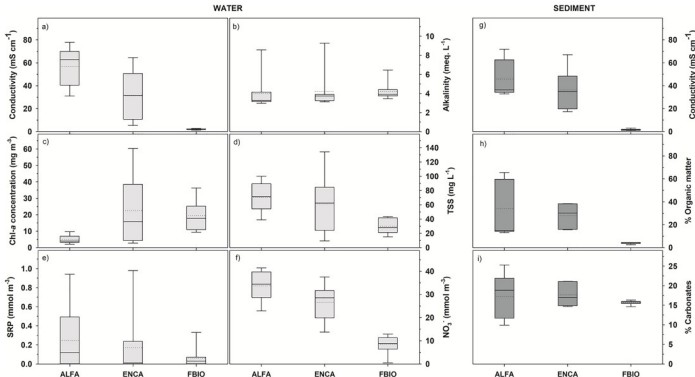

**Fig 2. Box-whisker plots of the values of the main physical and chemical variables in the three sampling sites.**
Dashed line—Mean, solid bold line—Median. For water parameters (a to f) n = 10, corresponding to 10 sampling dates. For sediment parameters (g to i) n = 9, corresponding to 3 sampling dates and 3 replicates each. a) water electrical conductivity (mS cm$^{-1}$); b) water alkalinity (meq. L$^{-1}$); c) Chl-*a* (mg m$^{-3}$); d) total suspended solids (mg L$^{-1}$); e) soluble reactive phosphorus (mmol m$^{-3}$); f) NO$_3^-$ concentration (mmol m$^{-3}$); g) electrical conductivity of sediments (mS cm$^{-1}$); h) organic matter content in the sediment (%); i) carbonate content in the sediment (%).

freshwater inputs, which can modify its natural salinity range. On the contrary, average conductivity in the freshwaters of FBIO were around 2.0 mS cm$^{-1}$, and never exceeded 2.7 mS cm$^{-1}$. pH values were around 8 on the three sites. Alkalinity did not differ as well between the three studied wetlands, with values around 4 meq. L$^{-1}$.

Other water variables, such as the Chl-*a*, the organic matter content, and the inorganic nutrient concentration, are indicative of the ecological status of the sites, and reflect the impacts to which they are subjected according to the anthropic pressures identified. Average Chl-*a* concentrations were much higher in ENCA (22.5 mg m$^{-3}$) and FBIO, (19.6 mg m$^{-3}$), than in ALFA (5 mg m$^{-3}$). Maximum values reached were 60.3 and 36.3 mg m$^{-3}$, respectively, for the brackish and freshwater wetlands, whereas they never exceeded 10 mg m$^{-3}$ for the ALFA salt marsh. Contrarily, total suspended solids (TSS) and NO$_3^-$ were higher in ALFA, with 70.3 mg L$^{-1}$ and 33.7 mmol m$^{-3}$, respectively, on average. These values were also relatively high in ENCA (61.4 mg TSS L$^{-1}$ and 26.6 mmol NO$_3^-$ m$^{-3}$). Meanwhile, FBIO showed, by far, the lowest values, with averages of 29.9 mg L$^{-1}$, for TSS and of 8.5 mmol m$^{-3}$ for NO$_3^-$. The differences in the ammonium concentrations among sites were less pronounced (Fig 2 and S1 Table).

The study of the sediment also revealed important differences between the studied sites, being the most relevant the electrical conductivity (Fig 2 and S2 Table). As also shown for water, the most saline site was ALFA, with a conductivity in the sediment of around 45.8 mS cm$^{-1}$, slightly lower than in water. ENCA presented a conductivity in sediment of around 36.2 mS cm$^{-1}$, and FBIO of around 1.7 mS cm$^{-1}$, all of them close to water values. There was a significant correlation between the conductivity in water and sediment (Pearson correlation coefficient of 0.747, n = 9, p = 0.021). The percentage of organic matter in the sediment (on a dry weight basis) was higher in ALFA, 34.2 ± 25.0%, the most saline site with less degradation rates and ENCA, 28.8 ± 10.9%, with frequent inputs during years, whereas this percentage was much lower in the restored freshwater wetland FBIO 3.9 ± 0.7% where part of the upper organic sediments were removed during restoration. The carbonate contribution to sediment dry weight did not differ among the studied sites, with average values around 16–17%.

The rates of the main C-related metabolisms were, on average, higher during the warmest months (Fig 3). The largest interannual differences were shown in ALFA, where all the studied

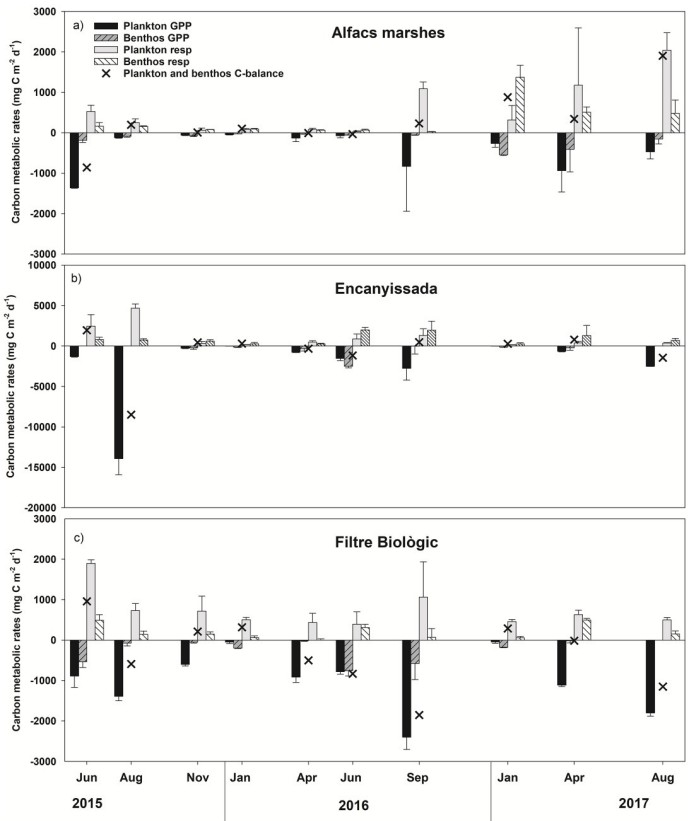

**Fig 3. Rates of the C-processes (Gross Primary Production–GPP-, and aerobic respiration) for plankton and benthos during the studied period in the three sampling sites, as well as the joint C-balance for plankton and benthos at each sampling date.** a) ALFA, b) ENCA, c) FBIO. n = 4 per sampling date and metabolic process. Negatives values mean a C-sink effect, and positive values mean a C-source effect. Narrow bars show the standard deviation. Note the differences in scales.

metabolisms were more active during the 2016–2017 cycle, and in ENCA, where these rates reached higher values in the two first sampling campaigns of the 2015–2016 cycle. Comparatively, the most productive site was ENCA, with average rates and peaks higher than those of the other sites. ALFA was the less productive system on average for all the studied metabolisms, although during the last sampling dates, some rates were higher than those of FBIO.

As shown in Fig 3a, metabolic rates in ALFA were generally weaker during 2015 and 2016, and higher in 2017. During the 2015–2016 cycle, plankton GPP rates were near or below 100 mg C m$^{-2}$ d$^{-1}$, but they reached 900 and 1,300 mg C m$^{-2}$ d$^{-1}$ in spring of 2017 and early summer of 2015. Plankton respiration rates in this site followed a similar pattern, although peaking higher, reaching 1,000 mg C m$^{-2}$ d$^{-1}$ in summer 2016 and spring 2017, and 2,000 mg C m$^{-2}$ d$^{-1}$ in summer 2017. The benthic compartment also showed rates that were, both for GPP and respiration, in the same magnitude order to those of the plankton compartment. The C-balance for plankton and benthos was mostly heterotrophic, even more remarkably in winter 2017 (+880 mg C m$^{-2}$ d$^{-1}$), and in summer 2017 (+1,900 mg C m$^{-2}$ d$^{-1}$). However, the balance was autotrophic in the early summer 2015 (-860 mg C m$^{-2}$ d$^{-1}$) and quite balanced both in winter 2015 and spring 2016.

In ENCA, metabolic rates were usually a magnitude order higher than in ALFA (Fig 3b). Plankton GPP reached 14,000 mg C m$^{-2}$ d$^{-1}$, the highest value obtained in our study, in late

summer 2015, but also overpassed 2,500 mg C m$^{-2}$ d$^{-1}$ in late summer 2016 and summer 2017. The highest plankton respiration rates generally coincided with the highest GPP values, although they were lower, 4,700 mg C m$^{-2}$ d$^{-1}$ in late summer 2015, and 1,300 mg C m$^{-2}$ d$^{-1}$ in late summer 2016. Only during both winters, plankton respiration was higher than plankton GPP. Benthos GPP showed lower rates on average than plankton, around 100–300 mg C m$^{-2}$ d$^{-1}$, although in some cases they were even below 20 mg C m$^{-2}$ d$^{-1}$. The highest benthic GPP peak of 2,500 mg C m$^{-2}$ d$^{-1}$ was found in the early summer 2016. Benthos respiration were above the GPP rates, never lower than 250 mg C m$^{-2}$ d$^{-1}$, and reaching 1,900 mg C m$^{-2}$ d$^{-1}$ both in early and late summer 2016. Jointly, the C-balance for plankton and benthos GPP and respiration was clearly autotrophic in late summer 2015 (-8,500 mg C m$^{-2}$ d$^{-1}$), spring and early summer 2016 (-330 and -1,170 mg C m$^{-2}$ d$^{-1}$ respectively), as well as in summer 2017 (-1,450 mg C m$^{-2}$ d$^{-1}$). Only for the rest of sampling dates the balance displayed a heterotrophic functioning, from +270 mg C m$^{-2}$ d$^{-1}$ in winter 2016 and 2017, to +2,000 mg C m$^{-2}$ d$^{-1}$ in early summer 2015.

In FBIO, plankton metabolic rates were higher than those of the benthos (Fig 3c). Plankton GPP was low in both winters studied, below 50 mg C m$^{-2}$ d$^{-1}$, meanwhile values reached 1,400, 2,400 and 1,800 mg C m$^{-2}$ d$^{-1}$, in late summer 2015 and 2016, and summer 2017, respectively. Plankton respiration rates were less variable and ranged between 400–1,000 mg C m$^{-2}$ d$^{-1}$, only surpassing this range in early summer 2015 (1,900 mg C m$^{-2}$ d$^{-1}$). Benthic GPP rates were usually below 200 mg C m$^{-2}$ d$^{-1}$, and only reached 500 mg C m$^{-2}$ d$^{-1}$ in early summer 2016, and early and late summer 2017. Benthos respiration rates were in all cases below 500 mg C m$^{-2}$ d$^{-1}$. Again, the net metabolic balance was heterotrophic during the coldest months when jointly considering both plankton and benthos, this is, in early and late winter 2016, and winter 2017, and also in early spring 2015. For the rest of the sampling dates, FBIO showed an autotrophic balance, with its maximum in late summer 2016 (-1,850 mg C m$^{-2}$ d$^{-1}$).

Helophytes and halophytes production followed a marked seasonal pattern, with production concentrated during the vegetative periods, spring and summer, and almost no production in winter (Fig 4). Higher biomass productions were recorded in FBIO, which was extensively populated by helophytes (*Phragmites australis*). Meanwhile, ENCA and ALFA, where due to their higher salinity helophytes were mostly replaced by halophytes (*Salicornia* spp.), generally showed lower production rates.

CH$_4$ emissions also followed a seasonal pattern, and its rates were higher in spring and summer, and very low in winter, (Fig 5). These rates were, on average, lower in ALFA, with values that barely reached 4–9 mg C m$^{-2}$ d$^{-1}$ in summer 2017 and early summer 2015, being almost negligible in winter. In ENCA, methane emissions peaked in early summer 2016 (162 mg C m$^{-2}$ d$^{-1}$), whereas the rest of moderately high rates were found in spring and summer

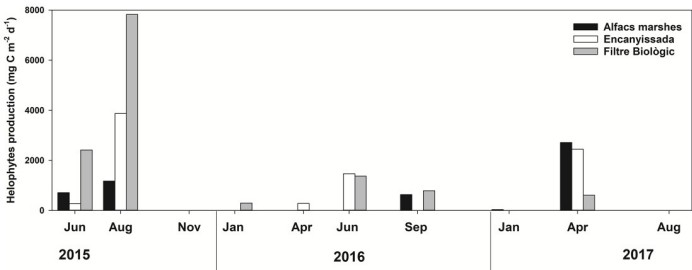

**Fig 4. Helophytes and halophytes net production rates for the three sampling sites.** n = 15 for the biomass estimation at each sampling date, from which C-capture is calculated.

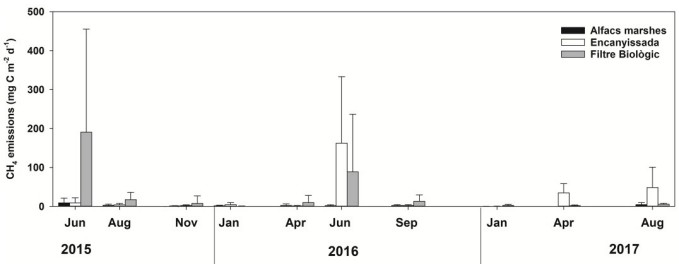

**Fig 5. CH₄ emission rates for the three sampling sites at each studied date.** n = 12 per site and sampling date. Narrow bars show the standard deviation.

2017 (4 and 48 mg C m$^{-2}$ d$^{-1}$, respectively). The site with highest methane emissions was FBIO, with maximum rates in early summer 2016 and 2017 (190 and 89 mg C m$^{-2}$ d$^{-1}$, respectively).

CH₄ emissions projected to expected environmental changes through experimental manipulations of temperature and salinity (Fig 6), showed a similar pattern for all studied sites, with exponential increases with increasing temperatures, and potential decreases with increasing conductivity. The highest significant correlations were found in ENCA for temperature (R = 0.78, p<0.0001) and in ALFA for both temperature (R = 0.78, p = 0.001) and conductivity (R = 0.79, p = 0.001). In all cases except for the conductivity in FBIO, correlations were significant (p<0.05). These experimental patterns confirmed the field data, where maximum rates were also found during the warmest months, as well as in the less saline system (FBIO). Conductivity ranged varied in ALFA, strongly influenced by the sea, from 30.3 to 78.8, where much lower values CH₄ emissions were achieved. In the case of ENCA and FBIO, being highly regulated systems and dependent on water flow regulation, hypothetically, low conductivity values could be more commonly achieved, which would trigger CH₄ emissions.

For a better understanding of the relationship between the environmental features and the metabolic rates, not only methane but also plankton and benthos GPP and respiration, and helophytes production, a multivariate analysis (Fig 7) was performed to show the main patterns. The first axis, explaining 64% of the variation, ordinated most FBIO samplings on its negative side, linked to helophytes production. Environmental variables associated were mainly the water depth and the amount of TSS and OM in water. The opposite side of axis 1, conductivity was the main featuring environmental factor, with benthic metabolisms, both GPP and respiration, mostly linked to ALFA samples. The second axis showed more clearly the effects of the trophic status, with ENCA samples clearly linked to chl-*a* and the plankton GPP.

Concerning the abundance of methanogens, the average relative abundance of methanogens reads regarding the total prokaryotic community in the sediments of the studied wetlands ranged 0.53–1.57% (Fig 8a). The maximum average values of relative abundance of methanogens were found in the ALFA sediment (1.57% ±0.42), being significantly higher than in ENCA (n = 3, p = 0.036) and in FBIO (n = 3, p = 0.041), while no significant (n = 3, p = 0.886) differences were found between the ENCA (0.53% ± 0.27) and the FBIO (0.55% ± 0.23) sediments. Up to 5 different classes of methanogens were found in the studied sediments. The class Methanobacteria and the order Methanomassiliicoccal (class Thermoplasmata) appeared in the sediment of all three wetlands, being *Methanomassiliicoccus* the most relevant genus. The class Methanomicrobia was found mainly in the sediments of FBIO, where up to three different orders of methanogens appeared (Methanocellales, Methanomicrobiales,

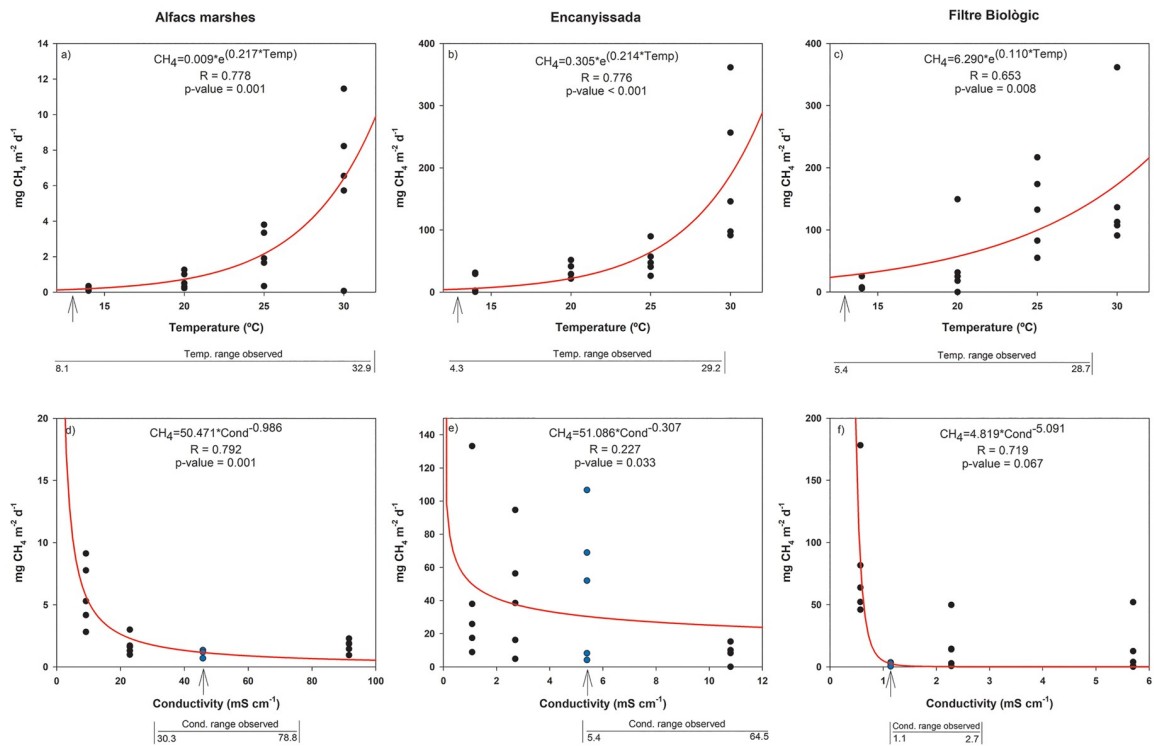

**Fig 6. Changes in CH$_4$ emission rates vs. temperature (above) and conductivity (below) for the studied wetlands.** The arrows show the water temperature and conductivity at the sites when samples for the experiments were obtained. The temperature and conductivity ranges registered during the 2 years-long study, respectively, are shown below each chart.

Methanosarcinales), with *Methanosphaerula*, *Methanosalsum*, *Methanosarcina* and *Methanothrix* being the most represented genera. The Methanococci class appeared mainly in the ENCA sediments, represented by the genera *Methanococcus* and *Methanothermococcus*. DNA-sequences corresponding to members of the class Methanopyri were found in ENCA and FBIO, represented by a single genus, *Methanopyrus*. Among the taxa found, Methanomassiliicoccales was the most abundant class in all sites representing 75.0 ± 3.3% of all the methanogens in ALFA, 66.5 ± 6.6% in ENCA and 55.2 ± 4.2% in FBIO. ANOVAs show that the only statistically significant difference among sites for Methanomassiliicoccales appeared between ALFA and FBIO (n = 3, p = 0.01), each on the opposite extremes of the salinity gradient.

The metabolic assignment results, performed with PICRUSt2, showed that the greatest metabolic capacity for acetoclastic methanogenesis was found in ALFA, followed by ENCA and FBIO (Fig 8c). This metabolic capacity was significant and positively correlated (regression analysis, n = 9, p = 0.008) to the organic matter content (%LOI) in the sediment (Fig 8b). However, although a positive trend was found between the relative contributions of total methanogens to the total prokaryotic metabolic capabilities and the organic matter content (%LOI) in the sediments, this was not demonstrated as significant at the 95% level (n = 9, p = 0.059). In contrast, a significant positive relationship between acetoclastic methanogenesis and the relative proportion of methanogens in the sediment was found (Fig 8c, n = 9, p = 0.049), showing that this type of methanogenesis was, by far, the dominating type in the studied sites.

Overall, the net metabolic C-balance was autotrophic in the three sampling sites, considering all C-metabolisms, namely the plankton and benthos GPP and respiration, the marginal

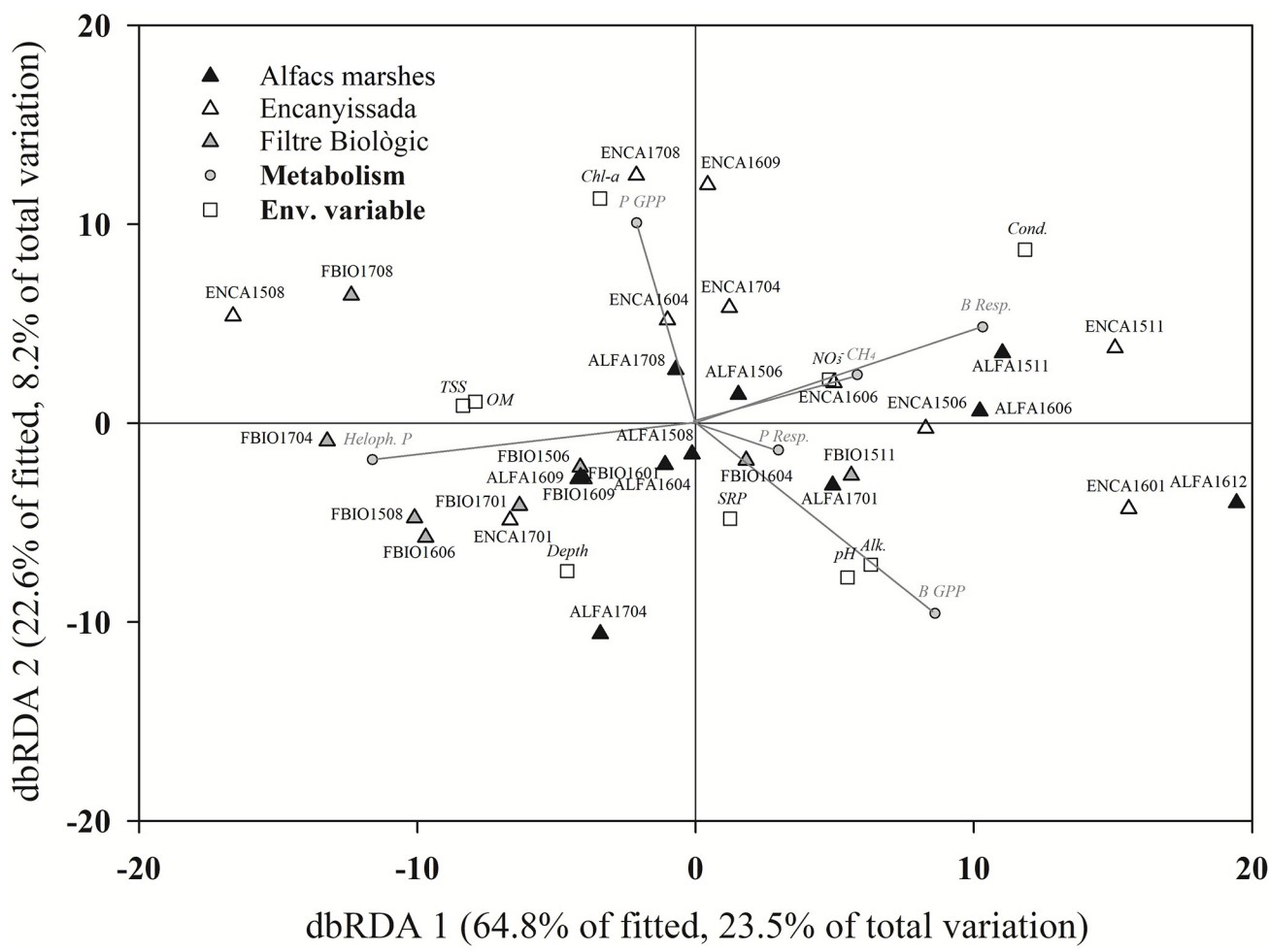

**Fig 7. Plot of a distance-based redundancy analysis (dbRDA).** Name of the samples refers to the date (NAMEYYMM). Environmental variables: Cond (conductivity), Depth (maximum water depth), pH, Alk (alkalinity), Chl-*a* (chlorophyll-*a* concentration), SRP (soluble reactive phosphorus), $NO_3^-$ ($NO_3^-$ concentration), OM (suspended organic matter), and TSS (total suspended solids). C-related processes: P GPP (plankton primary production), P Resp. (plankton respiration), B GPP (benthos primary production), B Resp. (benthos respiration), Heloph. P (helophytes production) and $CH_4$ (methane emissions).

vegetation production, and the $CH_4$ emissions (Fig 9a). Accordingly, C-sequestration capacity shown by the metabolic rates was around -42, -251 and -210 g C $m^{-2}$ $y^{-1}$, in ALFA, ENCA and FBIO, respectively. Plankton GPP predominated in ENCA, meanwhile helophytes production was dominant in FBIO, where plant production by *P. australis* production is higher when compared to the lower contribution of the halophytic *Salicornia* spp. in the more saline sites. $CH_4$ emissions were not relevant for the total C-balance, though the highest rates could remarkably influence the Global Warming Potential (GWP) balance. The GWP balance was also in the negative side for the three sampling sites, meaning the maintenance of a certain mitigating capacity (Fig 9b). However, when $CH_4$ emissions were more significant, especially in ENCA and FBIO, the GWP balance dropped the mitigation capacity in all cases to -29, -124 and -64 g $CO_2$-eq $m^{-2}$ $y^{-1}$ in ALFA, ENCA and FBIO, respectively.

When considering the total surface occupied by these wetland types in the Ebro Delta, (Fig 1 and Table 1), the total capacity of C-capture of these ecosystems in the Ebro Delta could

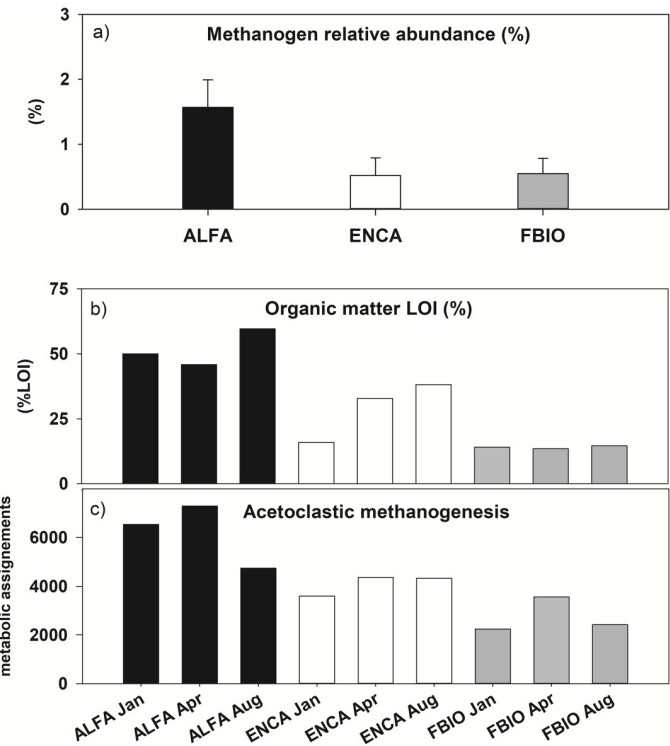

**Fig 8. a)** Percentage of methanogens abundance; **b)** organic matter content (LOI% in sediment, and **c)** metabolic assignments (from PICRUSt2) for the acetoclastic methanogens, for the three studied sites during three sampling events, fall winter, spring and summer 2017.

reach 6286 Tn C yr$^{-1}$, almost 3/4 of which would come from brackish wetlands, due to their higher C-sink behaviour (Fig 7) and the larger area they occupy.

## Discussion

This study contributes to the understanding of the role of deltaic wetlands in carbon flows and climate change mitigation, from the analysis of the influence of key environmental factors, such as salinity or trophic status, on carbon metabolisms and C-GHG emissions. Our results, showing the differences in the metabolic rates and microbial communities in different deltaic wetland types and conditions, can be used as a basis to define some conservation measures following a climatic approach as advances to reinforce the mitigating capacity of these wetlands.

Primary production and decomposition rates were in the same magnitude order than other studies done in similar deltaic wetlands in the Mediterranean region [27, 49]. Annual rates of C-retention were also in the rage of C-accretion rates in delta marshes found by Fennessy et al. [11] based on $^{137}$Cs (32–435 g C m$^{-2}$ yr$^{-1}$). However, these authors found higher C-accumulation in salt marshes, meanwhile our results suggested that coastal lagoons were slightly more productive. Salinity was demonstrated to be a major factor influencing C-metabolic rates and their balance [50–53]. As in our case, less saline sites showed higher metabolic rates when compared to those with higher salinities, which has implications in the C-sequestration [54]. Temperature is another main factor in determining the metabolic rates [24], and this determined the temporal patterns found in the three studied sites. Sediment features were also important for the C-metabolic rates and, although in our results did not show significant temporal variations, some spatial patterns were found. Sites with higher salinity contained a higher

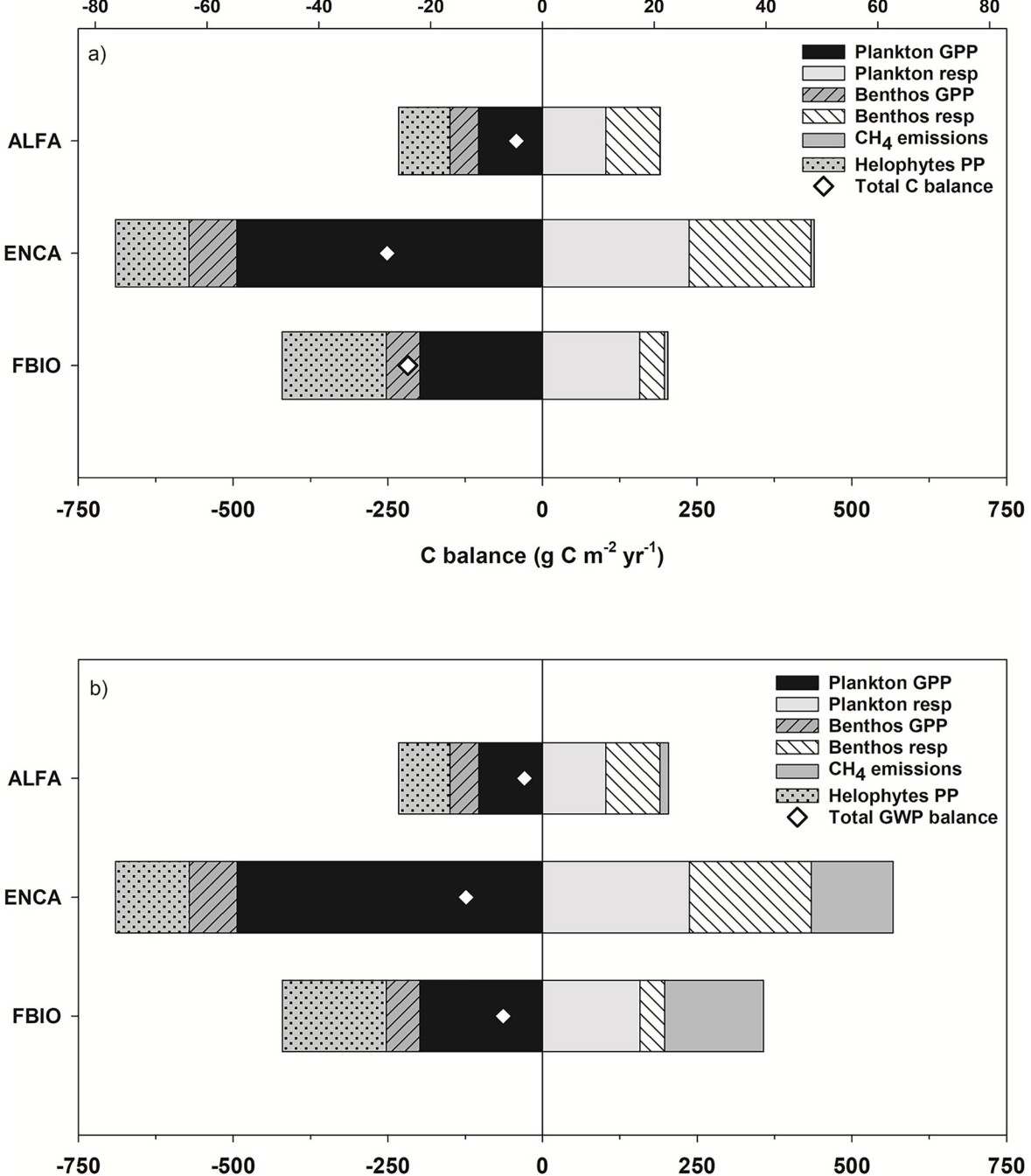

**Fig 9.** a). Annual C-balance (white diamond) and relative contribution to the C-balance of each of the C-processes studied in the three sampling sites. b) Annual GWP balance and relative contribution to the warming potential (or warming mitigation for C-capturing) of each of the C-processes, in g $CO_2$-eq m$^{-2}$ y$^{-1}$, obtained after extrapolation of bi-monthly rates. Negative values mean C-capture and mitigating capacity, whereas positive values mean C-release and warming capacity.

**Table 1. Total surface occupied by the different wetland types studied in the Ebro Delta, and extrapolation of the C-balance and GWP balance for these surfaces.**

| WETLAND TYPE | SURFACE (ha) | SITE STUDIED | C-balance (Tn C yr$^{-1}$) | GWP-balance (Tn CO$_2$-eq. yr$^{-1}$) |
|---|---|---|---|---|
| SALT MARSHES | 1292.43 | ALFA | -546.51 | -378.17 |
| BRACKISH COASTAL LAGOONS | 1797.40 | ENCA | -4514.50 | -2224.97 |
| FRESHWATER WETLANDS | 563.06 | FBIO | -1225.37 | -358.14 |

amount of organic matter in their sediments, probably because of their lower degradability rates allowing higher long-term accumulation, although removal processes during the restoration of FBIO explains its lower organic matter content. Organic matter content, at the same time, can influence some metabolic rates, especially respiratory processes in the benthic compartment [18], but also determine the composition of sediment microbial communities and the relative abundance of the different microbial guilds [26].

Other process determined by the salinity was the helophytes production. Emergent vegetation can dominate C-fixation, both in *P. australis* [18, 55] and *Salicornia* spp. dominated sites [51]. Our results were also supportive for this relevance, especially in the freshwater wetland, where an important part of the wetland surface was covered by *P. australis*, playing a prominent role in C-capture, as demonstrated by the multivariate analysis (Fig 7). A similar role is played by *Salicornia* spp. in more saline wetlands. This would support the proposal made by Rathore et al. [56], who argued on the resettlement of *Salicornia* for coastal soil rehabilitation in order to improve some ecological features. These restoration measures would, according to our data, greatly improve the C-capturing capacity.

As important as the ecological features for the metabolic C-balance, is the conservation status of the wetlands [18]. Carbon sequestration has been found to be negatively related with human disturbances [11]. The influence of some factors such as changes in salinity, and increases of trophic level, were already noted as huge threats for the deltaic wetland conservation in previous studies [49]. In this study, we showed how these changes in the studied wetlands influenced the metabolic rates and the C-balance, and how these effects could be reversed or, on the contrary, enhanced, when applying different conservation measures.

Freshwater inputs from the agriculture usually are heavily loaded with nutrients, causing eutrophication. In fact, nutrient concentration is one of the most important environmental factors influencing the metabolic rates and the total net ecosystem metabolism [25], which was also suggested previously for deltaic wetlands [31]. In our study, the most eutrophic system, ENCA, showing the poorest water quality, also presented the highest rates of plankton and benthos GPP and respiration, a main C-releasing process, and the multivariate also related these two features. According to the literature, eutrophication could enhance C-assimilation because of the increase of GPP rates [57–58]. However, this increase in GPP cannot be translated into a more burial efficiency, as remineralisation and degradative metabolisms are also enhanced [59]. CH$_4$ emissions can be particularly favoured by a higher organic matter production driving to the exhaustion of the most oxidised electron acceptors for the degradation of organic matter [58], thus increasing the warming capacity of wetland C-emissions. Therefore, another conservation action to be considered to reduce the rates of degradative metabolisms and C-emissions, thus increasing the mitigating effect would be the decrease of nutrient inputs.

One of the measures already implemented in the Ebro Delta is the (re)construction of wetlands acting as biological filters that reduce the nutrients loads from agricultural runoff, before entering the natural wetlands [30]. As demonstrated in our study, these wetland areas, now occupying abandoned rice fields, also show a high C-retention capacity and could be used for

the C-storage [60]. This kind of conservation solutions not only reduces the impacts on natural wetlands, but also generate many advantages like the increase of renaturalised areas [61], and enhance ecosystem services [15], which, according to our results, should now include also climate change mitigation as an important service of deltaic wetlands.

On the other side, variations in the natural salinity conditions of deltaic wetlands is another impact these wetlands, such as the selected studied sites, have to deal with. Combined effects of changes in land use and climate change would favour wetlands salinization [62]. Salinization can be accompanied by the eutrophication, and generates physical-chemical changes in water and consequences on biological communities [62]. Biogeochemical processes, such as $CH_4$ emissions or C-mineralisation, can be concomitantly altered [52]. Specifically, increases of salinity can bring down metabolic rates and decrease C-fixation [63], but also could increase the long-term storage capacity by depressing decomposition rates. According to our results, the site with a higher salinity, ALFA, showed in general lower metabolic rates. Moreover, higher salinity, especially when sulphate is widely available, can also be responsible of the decline of $CH_4$ production [24] by enhancing sulphate reduction [53], as we found in our study. Sulphate reducers outcompete methanogens since they have higher thermodynamic energy yield per unit of organic matter degraded, and turn $CH_4$ emissions to $CO_2$ emissions resulting from these anaerobic respiration processes, methanogenesis and sulphate reduction, respectively, with a much lower warming capacity at the short-medium term of the later [50]. Contrarily, increasing freshwater inputs in brackish coastal lagoons or salt marshes drives the ecosystem to a more heterotrophic behaviour [3]. According to our experiments, a reduction of the salinity can potentially result in increases of $CH_4$ emissions, as water salinity directly influences the sediment salinity with a significantly high correlation. Particularly, in ENCA this salinity fluctuations are highly relevant and show the human influence in this coastal lagoon that could alter the biogeochemistry of this wetland driving to increased methane emissions.

Being $CH_4$ emissions of minor importance in the carbon balance of the most saline wetland types studied, their implications on the global warming potential balance were also remarkable, as previously demonstrated for saline lakes [24]. In our study, the most saline sites were also characterized by higher organic matter concentrations in their sediments, which can be explained by its lower degradability under more saline conditions. A higher relative percentage of methanogens, as we found in ALFA, compared to less saline sites, did not drive to higher $CH_4$ emissions. That means that, although the relative relevance of metabolism assignments coming from the metabolic assignation made by PICRUSt2 could be indicative of the metabolic potential, current rates might be regulated by other factors. In this case, ALFA also displayed higher salinity and a high abundance of sulphate as being mostly seawater, which could counteract the relatively higher abundance of methanogens in the most saline site. However, these higher relative contributions of methanogens warn on the fact that they could be more active if some environmental changes, such as salinity decreases, as in this case $CH_4$ emissions could rise as demonstrated by our experiments. This would hardly happen in a salt marsh like ALFA, but it is more likely to occur in ENCA, where freshwater inputs are frequent, as a key in the management of this site.

Consequently, we showed that the relative abundance of microorganisms and its relative metabolic potential, versus the current activity expressed in the environmental conditions where these microorganisms are found, should be differentiated. For our specific case, the relative abundance of methanogens was significantly linked to the organic matter content of the sediment. This relative abundance would directly show that they are potentially capable of expressing methanogenesis, without necessarily being currently correlated with methane emissions. This is because the capacity of these methanogens for methane production can be

controlled by different factors, mainly temperature and salinity [24], the latter associated with the ability of sulphate reducers to outcompete methanogens in sulphate rich (as marine waters) saline environments [64]. Changes in these two main ecological factors, which can be associated to climate change but also to other components of the global change, such as changes in land uses, would potentially influence the rates of methanogenesis when occurring.

Conservation measures to maintain lower $CH_4$ emissions are essential to reduce the warming potential of these systems, by reducing freshwater inputs in salt marshes. Increase of organic matter in freshwater eutrophic sites, on the other hand, also drives to specific conditions that favour the methanogenesis [58–59]. $CH_4$ emissions in eutrophic systems and in constructed wetlands are also of highly relevance [65–66], although in FBIO, as part of the restoration plan, part of the organic sediment was removed, which explains the lower values of %LOI. Therefore, this and other sustainable measures should be considered for reducing their climatic impact [18].

Focusing on the microbial communities responsible of $CH_4$ emissions, members of the orders Methanobacteriales (class Methanobacteria) and Methanomassiliicoccales (class Thaumarchaeota) appeared in all the studied sites within the whole salinity gradient. Both classes, Methanobacteria and Thaumarchaeota have been related with high salinity environments. For instance, De Vrieze et al. [67] showed than in anaerobic digestion tanks, Methanobacteriales were selectively enriched as increasing salinity, and Guan et al. [68] showed that Thaumarchaeota were the most prevalent Archaea in oceanic deep hypersaline anoxic basins (DHABs). The wide tolerance of these organisms to salinity may allow their presence along the salinity gradient studied. Except for Methanosarcinales (class Metanomicrobia), the rest of methanogenic archaea in the sediments of the studied sites were quite scarce, all of them being unfavoured by a high salinity [69]. In our study, Methanosarcinales were more abundant in FBIO, although these archaea typically occur in anoxic sediments throughout the entire range of salinities [70].

Not only specific impacts, but other consequences of climate and global change are threatening deltaic ecosystems worldwide [71]. Ebro Delta is undergoing coastal retreat [17], associated to both the effects of climate change in the area and to the decreased sediment supply after damming most part of the lower Ebro river [28]. This led to a regression of coastal marshes, which receive much less sediment inputs because of the highly regulated river network [28]. With a reduction of the accretion rates, both carbon and mineral accumulation rates would also be reduced [11, 72]. Climate change, besides, could modify the structure and functioning of wetlands, altering their C-balance [13]. In the Ebro Delta, an increased sea level in areas that are mostly at less than 0.5 m above sea level, could have a huge effect. Episodes of severe storms, increasingly recurring in this area of the Mediterranean, threaten the Delta, and are currently impacting all types of wetlands, to the point of the loss of some of them, eaten by the advance of the sea. An impacting image of the possible effects is given by S1 Fig, a comparison of the Sentinel 1 radar images between January 15[th] and 21[st], before and after the storm developed in the eastern coast of the Iberian Peninsula, which shows how almost the entire delta is flooded, and sea has entered up to 3 km in the north part. Apart from the salinization/desalinisation, changes in the hydrological fluxes and morphological disturbances could alter the current disposition and structure of coastal wetlands [73], and the current microbial and vegetation communities could be replaced by others more adapted to these rapid environmental changes or less buffering conditions.

The wise way to preserve wetland habitats located in highly pressured landscapes, mainly by agricultural uses, like the Ebro Delta, would be through a better management and restoration planning [74]. Response of these ecosystems to predicted environmental changes should be considered in order to ensure the success of the implemented measures [70]. However, climate change is likely to have important implications for deltas restoration, and restoration

efforts will have to be more intensive to offset the impacts of climate change including accelerated sea level rise and changes in precipitation patterns [16]. Future coastal restoration efforts should also focus on nature-based solutions displaying techniques based on the own delta functioning for a sustainable conservation [19]. Management measures should consider freshwater and sediment flows from the river to balance negative effects from saline intrusion in freshwater wetlands, maintaining land elevation in threatened freshwater and brackish wetlands with river influences [75], but at the same time should avoid excess of irrigation leftovers entering the coastal salt marshes in order to maintain their natural structure and functioning, including its C-storage capacity.

Globally, the protection and restoration of coastal wetlands should consider the improvement of the C-sink functioning of wetlands [10, 20, 76]. The benefits associated with an appropriate wetland conservation and management for the C-mitigating capacity result mainly from two key ideas. First, a decrease of degradative metabolisms, both aerobic and anaerobic [18, 20], by a reduction of impacts and the return to natural ecological features, namely natural salinity level according to the ecological type, and trophic status. Second, the increase of C-retention capacity to move the balance towards a higher C-sink capacity and mitigating effect. Emergent marsh vegetation is more productive, and less labile, than submerged and floating vegetation [8], therefore, their relevant role on the wetland C-cycle can be of paramount importance for C-storage. When applying conservation and restoration measures, we should consider the capacity of ecosystems to increases C-removal from the atmosphere when restored and managed [77].

## Concluding remarks

As deltaic ecosystems are demonstrated to be highly active in the C-biogeochemistry and its link to climate change mitigation, their conservation could be defined also in terms that lead towards increasing the C-retention. Results obtained in this study suggest the influence of ecological features such as the salinity and the trophic status in the C-metabolic rates and can be determinant in the microbial activity that leads to GHG emissions. Knowing these mechanisms, a set of management and conservation measures can be established for the protection of these systems, strengthening their C-sink capacity, considering the implications of climate change and future predictions. The improvement of the natural wetlands structure and functions will enable to reduce degradative metabolisms in favour of the productivity and C-assimilation and storage on a medium/large term. Emergent plant communities, both helophytes and halophytes, can be helpful for the increase of C-retention. Control and reduction of impacts like freshwater inputs with high nutrients content, and salinization of freshwater and brackish systems, would favour the settlement of natural communities according to the sites characteristics, and the maintenance of the C-related metabolisms as climate allies. The construction of biological filters to reduce some of these impacts is also beneficial per se, as they can act as C-sinks, although more efforts have to be considered to reduce $CO_2$ and $CH_4$ emissions thereby. In the most saline marshes, contrarily, high salinity levels should be maintained, as in natural conditions, as they have potential methanogens in their sediments capable of activating methanogenesis when salinity levels considerably drop.

## Supporting information

**S1 Table. Main values and statistics of the water physical and chemical variables studied in each site.**
(DOCX)

**S2 Table. Main values and statistics of the sediment physical and chemical variables studied in each site.**
(DOCX)

**S1 Fig. Sentinel 1 radar images (2020-01-15 and 2020-01-22) of the Ebro Delta, north-eastern Spain, to show the comparison of the Delta before and after a storm, and the consequences of climate change in those ecosystems.** Source: Copernicus Services.
(TIF)

## Author Contributions

**Conceptualization:** Daniel Morant, Antonio Picazo, Carlos Rochera, Carles Ibañez, Maite Martínez-Eixarch, Antonio Camacho.

**Data curation:** Daniel Morant, Antonio Picazo, Carlos Rochera, Anna C. Santamans, Javier Miralles-Lorenzo.

**Formal analysis:** Daniel Morant, Antonio Picazo, Carlos Rochera, Antonio Camacho.

**Funding acquisition:** Antonio Camacho.

**Investigation:** Daniel Morant, Antonio Picazo, Carlos Rochera, Anna C. Santamans, Javier Miralles-Lorenzo, Alba Camacho-Santamans, Antonio Camacho.

**Methodology:** Daniel Morant, Antonio Picazo, Carlos Rochera, Antonio Camacho.

**Project administration:** Daniel Morant, Antonio Picazo, Carlos Rochera, Antonio Camacho.

**Resources:** Daniel Morant, Antonio Picazo, Carlos Rochera, Anna C. Santamans, Javier Miralles-Lorenzo, Alba Camacho-Santamans, Carles Ibañez, Maite Martínez-Eixarch, Antonio Camacho.

**Software:** Daniel Morant, Antonio Picazo, Carlos Rochera, Javier Miralles-Lorenzo.

**Supervision:** Antonio Camacho.

**Validation:** Daniel Morant, Antonio Camacho.

**Visualization:** Daniel Morant, Antonio Picazo, Carlos Rochera, Anna C. Santamans.

**Writing – original draft:** Daniel Morant.

**Writing – review & editing:** Antonio Camacho.

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
