## [Decision Letter · Decision Letter 0]

7 Jan 2020

PONE-D-19-34451

The role of ecological features and conservation status on the carbon cycle and methane emissions in the Ebro Delta wetlands

PLOS ONE

Dear Dr. Camacho,

Thank you for submitting your manuscript to PLOS ONE. After careful consideration, we feel that it has merit but does not fully meet PLOS ONE’s publication criteria as it currently stands. Therefore, we invite you to submit a revised version of the manuscript that addresses the points raised during the review process.

The study is interesting but has some problems as suggested by the reviewers. The authors should response to the comments of the reviewers one by one and revise the ms accordingly. The revised ms might be sent to the reviewers for further reviewing.

We would appreciate receiving your revised manuscript by Feb 21 2020 11:59PM. To enhance the reproducibility of your results, we recommend that if applicable you deposit your laboratory protocols in protocols.io, where a protocol can be assigned its own identifier (DOI) such that it can be cited independently in the future. For instructions see: http://journals.plos.org/plosone/s/submission-guidelines#loc-laboratory-protocols

We look forward to receiving your revised manuscript.

Kind regards,

Jian Liu

Academic Editor

PLOS ONE

Journal Requirements:

"This work was supported by the projects CLIMAWET (CGL2015-69557-R) and ECOLAKE (CGL2012-38909), funded by MINECO and FEDER-EU Funds, and the projects CARBONSINK and CARBONNAT funded by Fundación Biodiversidad, all of them awarded to AC. DM and JM-L hold a FPU Predoctoral Scholarship by the Spanish Ministry of Science, Innovation and Universities."

Please provide an amended Funding Statement that declares *all* the funding or sources of support received during this specific study (whether external or internal to your organization) as detailed online in our guide for authors at http://journals.plos.org/plosone/s/submit-nowPlease state what role the funders took in the study.  If any authors received a salary from any of your funders, please state which authors and which funder. If the funders had no role, please state: "The funders had no role in study design, data collection and analysis, decision to publish, or preparation of the manuscript."

4. We note that Figure 1 in your submission contains a map image which may be copyrighted.

We require you to either (a) present written permission from the copyright holder to publish this figure specifically under the CC BY 4.0 license, or (b) remove the figures from your submission:

b.    If you are unable to obtain permission from the original copyright holder to publish this figure under the CC BY 4.0 license or if the copyright holder’s requirements are incompatible with the CC BY 4.0 license, please either i) remove the figure or ii) supply a replacement figure that complies with the CC BY 4.0 license. Please check copyright information on all replacement figures and update the figure caption with source information. If applicable, please specify in the figure caption text when a figure is similar but not identical to the original image and is therefore for illustrative purposes only.

Reviewers' comments:

Reviewer's Responses to Questions

**Comments to the Author**

1. Is the manuscript technically sound, and do the data support the conclusions?

Reviewer #1: Partly

Reviewer #2: Yes

2. Has the statistical analysis been performed appropriately and rigorously? 

Reviewer #1: Yes

Reviewer #2: Yes

3. Have the authors made all data underlying the findings in their manuscript fully available?

Reviewer #1: Yes

Reviewer #2: Yes

4. Is the manuscript presented in an intelligible fashion and written in standard English?

Reviewer #1: Yes

Reviewer #2: Yes

5. Review Comments to the Author

Reviewer #1: Daniel Morant, et al. investigated the relationships between ecological features and conservation and carbon cycle and methane emissions in the Ebro Delta wetlands. The delta wetland plays an important role in carbon cycle and GHG emissions. Indeed, Daniel Morant et al. did a series of filed investigation and laboratory experiment in the Ebro Delta wetlands. However, I think that this paper still have some problems.

Problem 1, the title of this paper should be modified. Carbon cycle consists of many complicated processes, including methane emission. ‘Carbon cycle’ is not suitable for this paper. Both of ‘ecological features’ and ‘conservation status’ are ambiguous. Which features and conservation measures? This title is great. I think that it is not a good title for a paper, and is suitable for a scientific fund.

Problem 2, (1) the CO2 and CH4 emissions are always measured in situ, and the emission rate of CO2 and CH4 resulting from incubation experiment is unequal to the real rate in the field. Moreover, because of the temporal and spatial variation of CO2 and CH4 emissions, the field investigation of CO2 and CH4 emissions is always conducted more than two times a month (may be 3 times or more). However, the CH4 emission of this study is only measured in ten months from 2015 to 2017. And (2) the microbial community composition of sediment was tested using 16S rRNA technology. How its results support your investigation, I did not see them in the Abstract.

Problem 3, indeed, the authors did a series of experiment. Just like problem 1, it is impossible to clearly elaborate the processes of carbon cycle in a paper. Concerning a good paper, it is enough to clearly elaborate one scientific problem. In this paper, although many investigations were conducted, all the scientific problems were not clearly elaborated. What is the main scientific problem in your paper?

For example, (1) CO2 exchange in the wetland ecosystems is a complicated processes, and many factors adjust these processes, including plant, soil, hydrological conditions, soil microorganisms, and so on. The relationship between these processes and factors is still uncertain, especially the relationship between GHG emissions and the microbial community composition of soil or sediment. (2) CH4 emission also has a similar problem with CO2 exchange. And (3) how ecological conservation affects GHG emissions and its relationship with the microbial community composition of soil or sediment and the ecological features are still uncertain. I think that any of abovementioned scientific problems should be fully investigated, and must be a good paper.

Overall, this paper is not suitable for publication under this condition.

Reviewer #2: Comments

1) The authors mentioned that this survey covers two hydrological cycles. I would like to see more details about the hydrological changes in these wetlands, since they are all shallow and close to the sea and the river in the north. The monthly changes in precipitation and water level can be shown and I suppose the water level might be also one of the important factors affecting C cycle and methane emission. I noticed there are some sentences mentioning this but I don’t think it is adequate.

2) There are many minor typos then the ms needs carefully check. Here I list some of them, but not limited to these.

Line 142 “..”

Line 237 blank

Line 295 “’”in the last sentence

Line 347 “a”

Line 419 “.”

Line 425 “tall”

6. PLOS authors have the option to publish the peer review history of their article (what does this mean?). If published, this will include your full peer review and any attached files.

Reviewer #1: No

Reviewer #2: No

---

## [Author Response · Author response to Decision Letter 0]

4 Mar 2020

Dear Editor;

Below we detail the answers to the recommendations suggested, which are now reflected in the manuscript, both in the marked-up copy, as well as in the clean version of the revised paper with changes already accepted. 

Following, we address, point by point, all requests made.

Sincerely,

Antonio Camacho

To enhance the reproducibility of your results, we recommend that if applicable you deposit your laboratory protocols in protocols.io, where a protocol can be assigned its own identifier (DOI) such that it can be cited independently in the future. For instructions see: http://journals.plos.org/plosone/s/submission-guidelines#loc-laboratory-protocols

R: Although all methods we used are already published in our previous papers, which have been adequately referred, we have additionally created the drafts of the following specific protocols at protocols.io:

Estimation of gross primary production and aerobic respiration rates in the plankton and benthos in aquatic ecosystems

https://www.protocols.io/private/805543CB53D711EA916F0242AC110003

Estimation of methane emission rates in shallow lakes

https://www.protocols.io/private/EEC791A053BA11EA916F0242AC110003

R: Done.

2. In your Methods section, please provide additional information regarding the permits you obtained for the work. Please ensure you have included the full name of the authority that approved the field site access and, if no permits were

required, a brief statement explaining why.

R: Done, in lines 194-195 of the Manuscript document (all lines referred here to identify the changes made refer to the document with accepted changes.

R:Done.

4. We note that Figure 1 in your submission contains a map image which may be copyrighted.

R: The map is not copyrighted, as it is created by ourselves with free available shape (.shp) layers in a GIS software as explained below. We haven’t used any Google data neither copyrighted satellite images.

The map is created on free GIS software (Q-GIS) from different layers in shape formats, available for free from the following sources:

- Line and polygon layers from the Ebro river and basin were extracted from the downloadable cartography resources, Ministry for the Ecological Transition, Government of Spain (2019): https://www.miteco.gob.es/es/cartografia-y-sig/ide/descargas/agua/default.aspx

- Coastline of the Ebro Delta and the Iberian Peninsula were extracted from the downloadable polyline layer of Europe, European Environmental Agency (2019): https://www.eea.europa.eu/data-and-maps/data/eea-coastline-for-analysis-2/gis-data/eea-coastline-polyline

- Delineation and classification of wetlands used for the extrapolation were obtained from the C-LC maps, properly cited in the text, Copernicus (2019): https://land.copernicus.eu/pan-european/corine-land-cover

The other elements and the design were prepared by us.

On the other hand, we have included a Supplementary Figure made from Sentinel Images. The corresponding request for permission has already been sent to the Copernicus contact service. If we are granted (we should, as these are publicly usable images), we will keep that supplementary figure, otherwise it will be removed. 

5. Review Comments to the Author

Reviewer #1: Daniel Morant, et al. investigated the relationships between ecological features and conservation and carbon cycle and methane emissions in the Ebro Delta wetlands. The delta wetland plays an important role in carbon cycle and GHG emissions. Indeed, Daniel Morant et al. did a series of filed investigation and laboratory experiment in the Ebro Delta wetlands. However, I think that this paper still have some problems.

Problem 1, the title of this paper should be modified. Carbon cycle consists of many complicated processes, including methane emission. ‘Carbon cycle’ is not suitable for this paper. Both of ‘ecological features’ and ‘conservation status’ are ambiguous. Which features and conservation measures? This title is great. I think that it is not a good title for a paper, and is suitable for a scientific fund.

R: The tittle has been changed, as suggested, to “Carbon metabolic rates and GHG emissions in different wetland types of the Ebro Delta”. We now refer to C-metabolic rates or C-balances from a metabolic approach. GHG emissions are also explained from a metabolic point of view.

Regarding the ecological features and conservation status, we have focused only on salinity and trophic status, and the potential influences of changes in these two key factors on the C-metabolisms. A multivariate analysis, with the newly added Fig 7, shows how important are these two features for the wetlands status, as well as the relation of these and other physical and chemical parameters with the C metabolic rates: This was the main objective of the paper, providing scientific basis for the conservation measures suggested in order to increase the mitigating capacity.

Problem 2, 

(1) the CO2 and CH4 emissions are always measured in situ, and the emission rate of CO2 and CH4 resulting from incubation experiment is unequal to the real rate in the field. Moreover, because of the temporal and spatial variation of CO2 and CH4 emissions, the field investigation of CO2 and CH4 emissions is always conducted more than two times a month (may be 3 times or more). However, the CH4 emission of this study is only measured in ten months from 2015 to 2017. 

R: Sampling was designed as explained in the Ms, so these are the methods we start from, and the results we got. Knowing the spatial and temporal variability, our aim was to better differentiate among wetland types (in a salinity range) for conservation and climatic mitigation purposes, rather than studying a single site with more detail. Besides, as explained in the discussion, results are in line with other studies on carbon metabolic mechanisms or carbon balances in this type of systems.

Methane emissions results from field studies and from laboratory experiments are on the same magnitude order for the three selected sites under their natural conductivity and average field temperature. 

(2) And the microbial community composition of sediment was tested using 16S rRNA technology. How its results support your investigation, I did not see them in the Abstract.

R: This issue has been further developed. It has been included in the abstract (Lines 45-50), and better integrated in the Ms (results, Lines 510-548, and discussion, 679-727). Our results, which have been further shown and discussed, show the influence of the environmental factors, such as the sediment organic matter content on the abundance of methanogens, as well as the inhibition of methanogenesis by environmental features such as salinity. 

Problem 3, indeed, the authors did a series of experiment. Just like problem 1, it is impossible to clearly elaborate the processes of carbon cycle in a paper. Concerning a good paper, it is enough to clearly elaborate one scientific problem. In this paper, although many investigations were conducted, all the scientific problems were not clearly elaborated. What is the main scientific problem in your paper? 

For example, (1) CO2 exchange in the wetland ecosystems is a complicated processes, and many factors adjust these processes, including plant, soil, hydrological conditions, soil microorganisms, and so on. The relationship between these processes and factors is still uncertain, especially the relationship between GHG emissions and the microbial community composition of soil or sediment. (2) CH4 emission also has a similar problem with CO2 exchange. 

R: We reformulated and reoriented the objectives of the paper and have focused on two questions, portrayed in the results and discussion:

- How wetland salinity (as a key factor for the differentiation of types) and changes in the salinity as an impact) influence the C-metabolisms? And how methanogens are related to these metabolisms?

There is an inverse relationship between salinity and methane emissions (shown in the experiments, Fig 6, and discussed from Line 652 and so on). Trophic status also influences (see the multivariate analysis, Fig 7, as well as the discussion in Lines 619-642). Even though the most saline site has the lowest methane emission rates, it holds methanogens that keep the metabolic capacity in a sediment with more organic matter (discussed in Lines 688-705). This can have implications in the C and GWP balances if salinity changes due to impacts or management measures (Lines 706-713)

- How alterations of the trophic status influence C-metabolisms and C-GHG?

As now said, for plankton and benthos metabolisms, a increased trophic status increases both assimilative and degradative C-metabolisms (Lines 619-642). Its implications for the C and GWP balances has now been discussed.

Responses to these questions, apart from the in situ and experimental studies, are also seen in the newly added multivariate analysis (new Fig 7), where the relationship of these metabolisms with environmental factors is shown. 

And (3) how ecological conservation affects GHG emissions and its relationship with the microbial community composition of soil or sediment and the ecological features are still uncertain. I think that any of abovementioned scientific problems should be fully investigated, and must be a good paper. 

R: GHG emissions and the microbial community are mainly studied in relation to the salinity. Changes in the salinity can be seen as an impact factor with negative consequences for the mitigating capacity, as explained in the Ms (Results, see Fig 3, 4 and 5, discussion, Lines 676-692). Ecological conservation, referred to changes in the salinity and trophic status, as well as other physical and chemical values, are seen in relation to metabolic rates and the consequent C-balances in the multivariate analysis. This multivariate analysis clearly ordinates the samples, the environmental variables, and the main C-metabolisms, showing the pattern by which they are related.

Overall, this paper is not suitable for publication under this condition.

Reviewer #2: Comments

1) The authors mentioned that this survey covers two hydrological cycles. I would like to see more details about the hydrological changes in these wetlands, since they are all shallow and close to the sea and the river in the north. The monthly changes in precipitation and water level can be shown and I suppose the water level might be also one of the important factors affecting C cycle and methane emission. I noticed there are some sentences mentioning this but I don’t think it is adequate.

R: Being highly regulated wetlands with surrounding rice fields and a network of channels to control the irrigation water, the coastal brackish lagoon and freshwater wetland are also regulated and barely show variation in depth (now this explanation has been added in Lines 170-171 and 183-184). See also Supplementary Table 1, where the data about the depth is shown. The other site, Alfacs, is a microtidal salt marsh (as explained in Line 165), whose depth is more related to the sea level rather than to rainfall precipitation. This is the reason why we did not go deeper into this topic. 

2) There are many minor typos then the ms needs carefully check. Here I list some of them, but not limited to these.

Line 142 “..”

Line 237 blank

Line 295 “’”in the last sentence

Line 347 “a”

Line 419 “.”

Line 425 “tall”

R: The text has been deeply reviewed to correct these and other minor misspellings.

---

## [Decision Letter · Decision Letter 1]

31 Mar 2020

Carbon metabolic rates and GHG emissions in different wetland types of the Ebro Delta

PONE-D-19-34451R1

Dear Dr. Camacho,

We are pleased to inform you that your manuscript has been judged scientifically suitable for publication and will be formally accepted for publication once it complies with all outstanding technical requirements.

With kind regards,

Jian Liu

Academic Editor

PLOS ONE

Additional Editor Comments (optional):

Reviewers' comments:

Reviewer's Responses to Questions

**Comments to the Author**

1. If the authors have adequately addressed your comments raised in a previous round of review and you feel that this manuscript is now acceptable for publication, you may indicate that here to bypass the “Comments to the Author” section, enter your conflict of interest statement in the “Confidential to Editor” section, and submit your "Accept" recommendation.

Reviewer #2: All comments have been addressed

2. Is the manuscript technically sound, and do the data support the conclusions?

Reviewer #2: Yes

3. Has the statistical analysis been performed appropriately and rigorously? 

Reviewer #2: Yes

4. Have the authors made all data underlying the findings in their manuscript fully available?

Reviewer #2: Yes

5. Is the manuscript presented in an intelligible fashion and written in standard English?

Reviewer #2: Yes

6. Review Comments to the Author

Reviewer #2: (No Response)

7. PLOS authors have the option to publish the peer review history of their article (what does this mean?). If published, this will include your full peer review and any attached files.

Reviewer #2: No

---

## [Editor Report · Acceptance letter]

9 Apr 2020

PONE-D-19-34451R1 

Carbon metabolic rates and GHG emissions in different wetland types of the Ebro Delta 

Dear Dr. Camacho:

I am pleased to inform you that your manuscript has been deemed suitable for publication in PLOS ONE. Congratulations! Your manuscript is now with our production department. 

With kind regards,

on behalf of

Dr. Jian Liu 

Academic Editor

PLOS ONE